# Implication of thermal signaling in neuronal differentiation revealed by manipulation and measurement of intracellular temperature

Shunsuke Chuma[1,2], Kazuyuki Kiyosue [3], Taishu Akiyama[4,5], Masaki Kinoshita [4,5], Yukiho Shimazaki[1,2], Seiichi Uchiyama [6], Shingo Sotoma[1], Kohki Okabe [6,7] ✉ & Yoshie Harada [1,5,8,9] ✉

Neuronal differentiation—the development of neurons from neural stem cells—involves neurite outgrowth and is a key process during the development and regeneration of neural functions. In addition to various chemical signaling mechanisms, it has been suggested that thermal stimuli induce neuronal differentiation. However, the function of physiological subcellular thermogenesis during neuronal differentiation remains unknown. Here we create methods to manipulate and observe local intracellular temperature, and investigate the effects of noninvasive temperature changes on neuronal differentiation using neuron-like PC12 cells. Using quantitative heating with an infrared laser, we find an increase in local temperature (especially in the nucleus) facilitates neurite outgrowth. Intracellular thermometry reveals that neuronal differentiation is accompanied by intracellular thermogenesis associated with transcription and translation. Suppression of intracellular temperature increase during neuronal differentiation inhibits neurite outgrowth. Furthermore, spontaneous intracellular temperature elevation is involved in neurite outgrowth of primary mouse cortical neurons. These results offer a model for understanding neuronal differentiation induced by intracellular thermal signaling.

Neurons are cells that differentiate from neural stem cells[1]. They undergo dramatic morphological changes during differentiation and subsequent acquisition of neural functions[2,3]. Neural differentiation is essential to the process by which brain functions form and is also associated with the regeneration of neural function, which has potential applications in the treatment of various brain diseases and damage. Therefore, the complicated mechanism of neural differentiation has long been the focus of intensive study[4–6].

Previous efforts have revealed complex molecular mechanisms underlying neuronal differentiation. For example, it is induced by extracellular factors such as neurotrophins, represented by nerve growth factor (NGF), which activates various chemical signaling

[1]Institute for Protein Research, Osaka University, 3-2 Yamadaoka, Suita, Osaka 565-0871, Japan. [2]Graduate School of Science, Osaka University, 1-1 Machikaneyamacho, Toyonaka, Osaka 560-0043, Japan. [3]Biomedical Research Institute, National Institute of Advanced Industrial Science and Technology (AIST), 1-8-31 Midorigaoka, Ikeda, Osaka 563-8577, Japan. [4]Graduate School of Biostudies, Kyoto University, Yoshida-Konoecho, Sakyo-Ku, Kyoto, Kyoto 606-8501, Japan. [5]Institute for Integrated Cell-Material Sciences (WPI-iCeMS), Kyoto University, Yoshida-Honmachi, Sakyo-Ku, Kyoto, Kyoto 606-8501, Japan. [6]Graduate School of Pharmaceutical Sciences, The University of Tokyo, 7-3-1 Hongo, Bunkyo-Ku, Tokyo 113-0033, Japan. [7]JST, PRESTO, 4-8-1 Honcho, Kawaguchi, Saitama 332-0012, Japan. [8]Center for Quantum Information and Quantum Biology, Osaka University, 1-2 Machikaneyamacho, Toyonaka, Osaka 560-0043, Japan. [9]Premium Research Institute for Human Metaverse Medicine (WPI-PRIMe), Osaka University, 2-2 Yamadaoka, Suita, Osaka 565-0871, Japan. ✉e-mail: okabe@mol.f.u-tokyo.ac.jp; yharada@protein.osaka-u.ac.jp

pathways. Activation of tropomyosin receptor kinase A (TrkA), a high-affinity NGF receptor, stimulates the rapidly accelerated fibrosarcoma (RAF)/mitogen-activated protein kinase 1/2 (MEK1/2)/extracellular signal-regulated kinase 1/2 (ERK1/2) signaling pathway and activates transcription[7]. Additionally, TrkA activation has multiple cellular effects, including translation activation, through the phosphoinositide 3-kinase (PI3K)-mediated AKT pathway[8]. The low-affinity NGF receptor p75 induces transcription via nuclear factor kappa B (NF-kB). Thus, these signaling pathways activate transcription and translation in differentiating neural stem cells, leading to actin polymerization[9,10], microtubule synthesis[11], and, consequently, neurite outgrowth[12,13].

Physical signals such as mechanical force[14,15], electric field[16,17], light[18–21], and temperature[22–29], have also been reported to influence neuronal differentiation. In particular, spontaneous temperature fluctuations, which are larger than the changes in deep body temperature (<1 °C) due to biological rhythms (e.g., circadian rhythm[30] or menstrual cycle[31]), occur in the brain[32,33]. These fluctuations suggest a profound relationship between temperature and brain functions. Recently, the relationship between neurite outgrowth and temperature has been investigated, showing that thermal stimuli (e.g., heat shock) induce neurite outgrowth[23–25,28]. Additionally, the direction of neurite outgrowth (i.e., axon guidance) is influenced by extracellular heating[26,27]. These studies on the induction of neuronal differentiation by thermal stimulation involved mainly manipulating the temperature of the entire culture medium or the extracellular vicinity. In these methods, either the entire cell is heated or the heat source is located outside the cell; thus, it is not possible to understand the cellular response to local temperature changes (e.g., in the nucleus or the cytoplasm) or the cellular-intrinsic local temperature changes during differentiation. As such, the mechanism of neuronal differentiation involving intracellular temperature remains unknown.

Advances in intracellular temperature measurement techniques, however, have led to descriptions of cellular-intrinsic, location-dependent temperature variations, such as in the nucleus and the cytoplasm of individual cells[34–37]. Moreover, stimuli-responsive intracellular temperature variations in brain tissue[38] and neurons[39,40] have also been associated with their respective functions. This implies that local intracellular temperature changes are likely important for the temperature-related regulation of neuronal differentiation as a thermal signaling mechanism[41]. Indeed, the regulation of cellular functions by noninvasive intracellular heating as quantified by intracellular thermometry techniques has been reported, including in neuronal functions[42–44].

In this study, we focused on NGF-dependent neurite outgrowth in neuron-like PC12 cells to investigate the effects of an increase in temperature on this neuronal differentiation phenomenon. First, we heated PC12 cells locally and quantitatively using infrared (IR) laser irradiation to show that the temperature increase in the nucleus promotes neurite outgrowth. Second, using two different intracellular thermometers, we observed a spontaneous temperature increase dependent upon nuclear transcription and cytoplasmic translation during neuronal differentiation. Based on these findings, we offer a unique model of a thermal signaling-mediated neuronal differentiation mechanism in which a spontaneous temperature increase contributes to neurite outgrowth.

## Results

### Investigation of thermal signaling in neuron-like PC12 cells using fluorescent thermometers

PC12 cells differentiated into neuronal-like cells with elongated neurites upon NGF treatment (Fig. 1a). In this study, we defined differentiated cells as those with neurites longer than the length of the cell body. Neurite length was measured as the length of the longest protrusion in differentiated cells as opposed to 0 μm observed in undifferentiated cells. Neurite outgrowth occurred approximately after 6 h

of NGF treatment. The average neurite length reached approximately 71 μm after 24 h (Fig. 1b). This differentiation phenomenon occurred in 81% of cells observed. In conducting a quantitative evaluation of the neurite outgrowth efficiency by measuring protrusion length and differentiation rate, we focused on protrusion length as the primary index for evaluation. We examined the expression of a marker of neuronal differentiation (microtubule-associated protein 2, MAP2) before and after the addition of NGF to confirm the acquisition of differentiated phenotypes. Immunostaining of cells confirmed the expression of MAP2 in NGF-treated cells (Supplementary Fig. 1). Next, we investigated the involvement of intracellular temperature in neuronal differentiation using intracellular thermometry and local heating (Fig. 1c). For local (φ < 5 μm) intracellular heating, we used IR laser (1475 nm) irradiation[45]. To capture physiological temperature variations, we employed fluorescent polymeric thermometers (FPTs) with high sensitivity (Fig. 1d and Supplementary Note 1)[34,37] and fluorescent nanodiamonds (FNDs) with high robustness[46,47] (Fig. 1e and Supplementary Note 1). In addition to the conventional FPT[34] (hereafter referred to as $FPT_{Low}$), we developed $FPT_{High}$, which responds to a higher temperature range (Supplementary Fig. 2). The FPTs and FNDs used in this study showed temperature-dependent changes in fluorescence lifetime (Fig. 1f and Supplementary Fig. 3) and optically detected magnetic resonance (ODMR, Fig. 1g and Supplementary Fig. 4), respectively.

### Quantification of localized intracellular heating induced by IR laser using intracellular thermometry

Intracellular temperature mapping using $FPT_{High}$ during localized intracellular heating by an IR laser allowed us to quantify the temperature changes and spatial distribution (i.e., temperature gradient) in PC12 cells in response to various irradiation intensities (Fig. 2a). The results showed that a regionally limited temperature gradient of <5 μm was formed in IR laser-irradiated cells (Fig. 2b). The extent of temperature change and the temperature gradient in response to IR laser irradiation of various intensities were examined in the nucleus and cytoplasm. An artificial temperature gradient could be generated locally in these areas (Fig. 2c). Estimation of the average temperature change in the areas where the temperature increased due to heating confirmed the heating capability of the IR laser in the nucleus and cytoplasm in an intensity-dependent manner (Fig. 2d). The standard error of the quantified temperature change ($\Delta T$) was approximately 10% (e.g., $\Delta T = 3 \pm 0.3$ °C for 150 μW heating). We hypothesize that the variation in temperature change ($\Delta T$) may be due to the potential differences in thermodynamic environments between cells or between local regions within cells (i.e., cells or regions that are easily warmed and regions that are not). As the heated region in this study is small (5 μm), these local environmental effects are unlikely to be averaged out.

### Promotion of neurite outgrowth by localized intracellular heating during NGF-mediated neuronal differentiation

To evaluate the effects of local temperature increase in cells on neuronal differentiation, local and quantitative heating in single cells was performed using an IR laser for a certain time during neuronal differentiation of NGF-treated PC12 cells. Neurite length and formation rate (defined as the percentage of cells that formed neurites longer than the cell body) were measured after 24 h of NGF treatment (Fig. 3a). To account for the effects of different experimental conditions (e.g., incubation time at the microscope) during heating on neurite outgrowth, control experiments without stimulation were performed for each experimental condition. We chose the following timing of heating: immediately (within 30 min, condition [i]) and 6 h (just before the initiation of projection elongation, condition [ii]) after the addition of NGF (Fig. 1b). For observations, target cells were visualized by microinjection of Texas Red-labeled dextran (TR-Dex) immediately after the addition of NGF. Compared with unheated cells, cells in which the

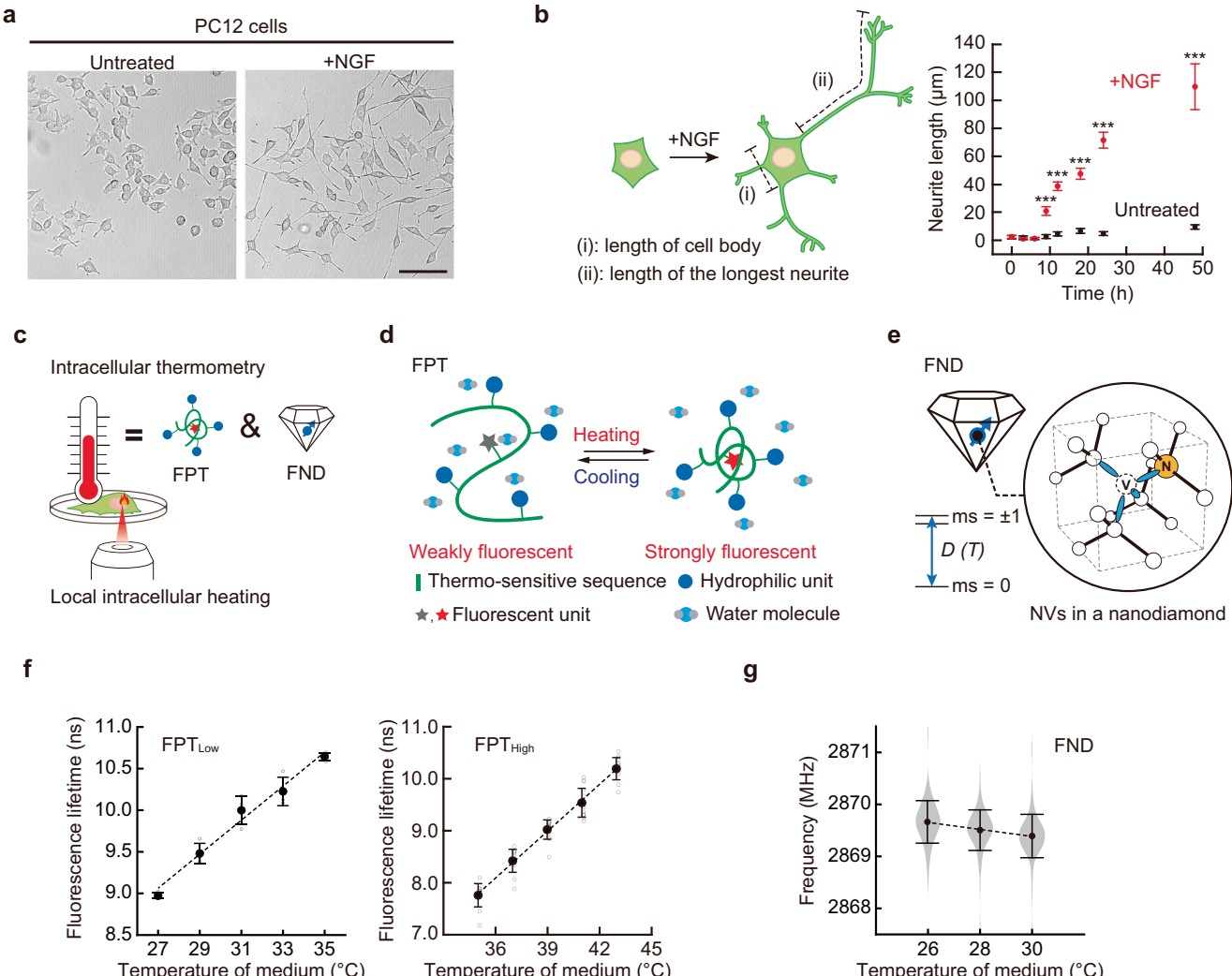

Fig. 1 | **Investigation of the role of intracellular temperature in neuronal differentiation using PC12 cells. a** Induction of neuronal differentiation by nerve growth factor (NGF). Like in neurons, induction of differentiation caused neurite outgrowth in PC12 cells. The scale bar represents 100 μm. **b** Changes in neurite length during neuronal differentiation induced by NGF are shown in (**a**). Neurites were defined as projections (ii) longer than the cell body (i). Time-course analysis of neurite growth (right). Data were shown as means ± standard errors of means (s.e.m.) ($n = 80$ [0, 3, 6, 9, 18, 24 h], 100 [12 h], 101 [48 h] cells for untreated and $n = 80$ [0, 3, 6 h], 100 [9, 12, 24, 48 h], 93 [18 h] cells for +NGF over two independent experiments). ***$P < 0.001$ (one-sided unpaired Student's $t$-test). **c** Local intracellular heating using IR laser and intracellular temperature measurement using fluorescent polymeric thermometers (FPTs) and fluorescent nanodiamonds (FNDs). **d** Principle of thermometry with FPTs. **e** Principle of thermometry with FNDs. The structure of

nitrogen-vacancy centers (NVC, enlarged) in a diamond crystal is shown. White circles, N, and V represent carbon atoms, nitrogen atoms, and vacancy, respectively. The NVC has two electronic spin states in its orbital ground state, ms = 0, ±1, which are energetically separated by a temperature-dependent, zero-field splitting $D$ (T). **f** Temperature-dependent changes in fluorescence lifetime of FPTs. Data were shown as means ± standard deviations (s.d.) ($n = 4$ [27, 29, 31, 33 °C], 5 [35 °C] cells for FPT$_{Low}$ over two independent experiments and $n = 12$ [27, 35 °C], 13 [29 °C], 14 [31, 33 °C] cells for FPT$_{High}$ over three independent experiments). The dotted lines represent linear fitting ($R^2 = 0.99$ for FPT$_{Low}$ and 0.99 for FPT$_{High}$). **g** Temperature-dependent change in resonance frequency of $D$ (T) of FNDs in a medium. Data were shown as means ± s.d. ($n = 77$ [26 °C], 81 [28, 30 °C] particles over two independent experiments) with violin plot. The dotted line indicates linear fitting ($R^2 = 0.99$). Source data are provided as a Source Data file.

nucleus was heated for 1 h immediately after NGF addition (condition [i]) had longer neurites after 24 h of NGF treatment (Fig. 3b). To investigate the effect of heating intensity on neurite outgrowth, we measured neurite outgrowth for different temperature differences ($\Delta T = +2, 3$, and 5 °C) and fixed heating time (1 h). Neurites were longer depending on the applied temperature increment (Fig. 3c and Supplementary Fig. 5a). Some cells had detached from the dish during or after heating at high laser intensity ($\Delta T = +5$ °C) (Supplementary Fig. 6). To avoid this heat-induced cytotoxicity, a milder heating intensity ($\Delta T = +3$ °C) was therefore chosen optimally.

Next, we examine the effects of heating time on neurite outgrowth using different heating durations (10, 20, 30, and 60 min) and a fixed heating intensity condition ($\Delta T = +3$ °C). Heating for 10 min did not affect neurite outgrowth, whereas heating for 20 min

or longer significantly promoted neurite outgrowth. Heating for 30 min had the strongest effect on neurite outgrowth (Fig. 3d and Supplementary Fig. 5b). The same heating conditions ($\Delta T = +3$ °C, 30 min) applied after 6 h of NGF treatment (condition [ii]) resulted in a moderate enhancement of neurite outgrowth which was weaker than that observed when heating immediately after the addition of NGF (Fig. 3e and Supplementary Fig. 5c). These results indicated that a certain increase in temperature in the nucleus for a certain amount of time immediately after the addition of NGF promotes neurite outgrowth.

Likewise, we examined the effect of cytoplasmic heating on neuronal differentiation, which showed that its impact on the promotion of neurite outgrowth was weaker than that of nuclear heating (Fig. 3f, g, Supplementary Fig. 5d–f, and Supplementary Note 2).

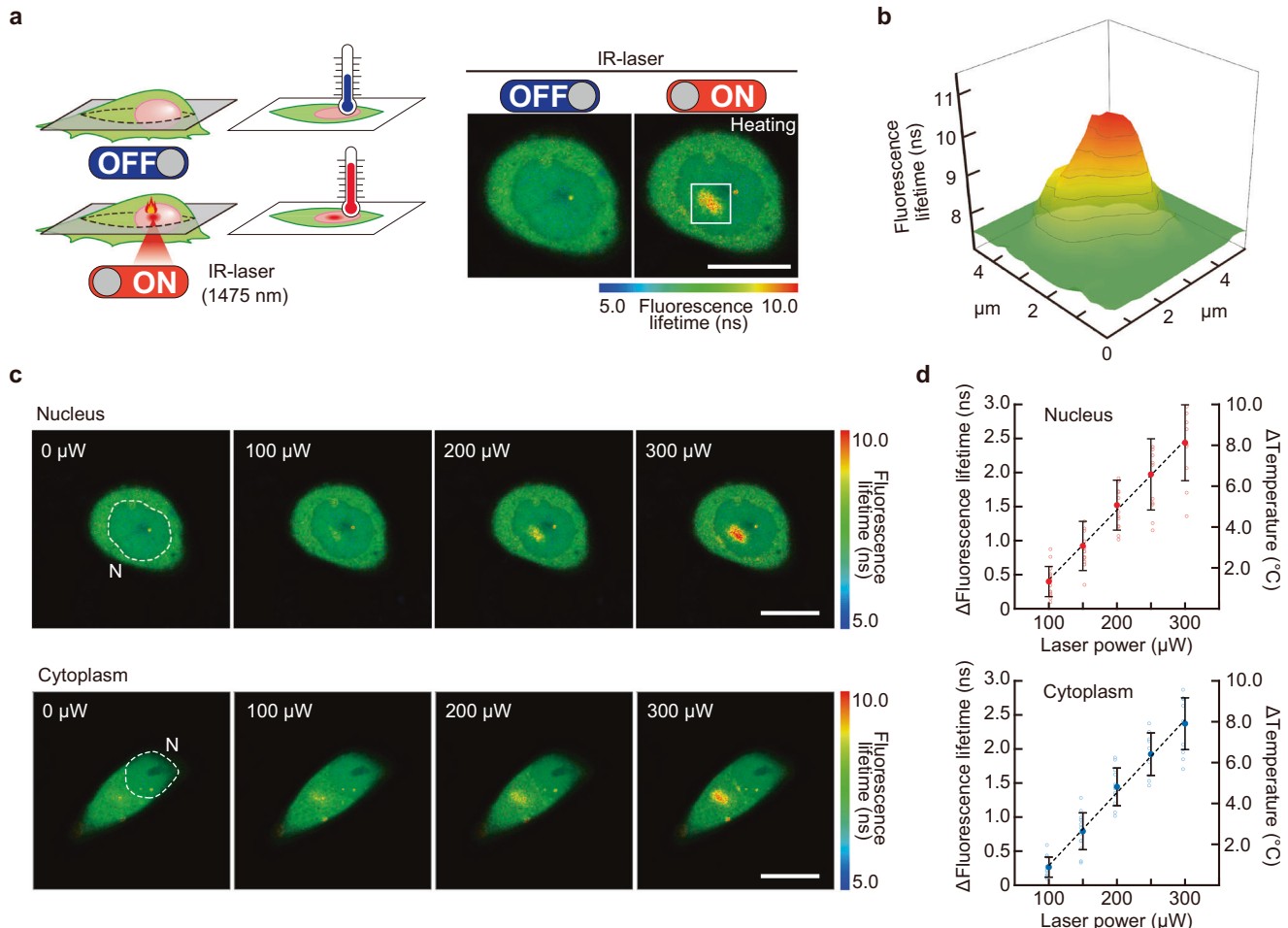

**Fig. 2 | Quantitative local heating in living PC12 cells using IR laser irradiation.** **a** Schematic and representative confocal fluorescence lifetime images of intracellular heating (in the nucleus) using infrared (IR) laser irradiation (250 μW). The scale bar represents 10 μm. **b** Temperature gradient induced by heating with IR laser irradiation in (**a**). **c** Changes in temperature distribution in the nucleus or cytoplasm due to IR laser irradiation at various intensities. Scale bars represent 10 μm. N indicates the nucleus. **d** Calibration curves of fluorescence lifetime and corresponding temperature increase in irradiated areas shown in (**a**–**c**) as a function of IR laser power. Data were presented as means ± s.d. ($n = 14$ [100, 150 μW], 13 [200 μW], 12 [250 μW], 10 [300 μW] cells for heating in the nucleus and $n = 11$ [100, 200 μW], 13 [150 μW], 9 [250 μW], 12 [300 μW] cells for heating in the cytoplasm over three independent experiments). Dotted lines indicate linear fitting ($R^2 = 0.99$ for the nucleus and cytoplasm). Source data are provided as a Source Data file.

Furthermore, MAP2 expression was increased in NGF-treated cells in which local regions of the nucleus were heated under the same conditions ($\Delta T = +3\,°C$, 30 min) as compared to cells that were not heated (Supplementary Fig. 7).

## Induction of neurite outgrowth by localized intracellular heating in the absence of NGF

To determine whether a temperature increase in the nucleus is a factor inducing differentiation, we investigated whether intracellular heating without treatment with NGF, which activates various chemical signaling pathways, induces neurite outgrowth in PC12 cells (Fig. 4a). After local nuclear heating ($\Delta T = +3\,°C$, 30 min), cells were cultured for 24 h without NGF. Then, neurite outgrowth was analyzed (Fig. 4b). Local heating of the nucleus for 30 min significantly increased the percentage of cells with neurite outgrowth (Fig. 4c, d). These results suggest that a local increase in temperature in the nucleus directly induces neuronal differentiation. These effects of heating in the absence of NGF suggest that the local temperature increase in the nucleus itself functions as a trigger for neuronal differentiation. This may be activated via a change at the molecular level and in its complex state (etc.) rather than via an acceleration of an already ongoing intracellular reaction due to an increase in temperature.

## Intracellular temperature increases during neuronal differentiation

In recent years, spontaneous intracellular temperature changes have been associated with brain/neuronal functions[38–40]; therefore, we investigated whether spontaneous temperature changes occurred in PC12 cells during the process of neuronal differentiation. Intracellular temperatures were measured before and after neuronal differentiation using FPTs and FNDs (Fig. 1c). Intracellular temperature imaging using $FPT_{Low}$ showed that in predifferentiated PC12 cells, the nuclear temperature was higher than that of the cytoplasm, which was consistent with previous studies[34,35,41]. In cells differentiated upon addition of NGF, the intracellular temperature was increased throughout the cell (Fig. 5a, b and Supplementary Fig. 8). Using a calibration curve (Fig. 1f) to calculate the average temperature change in intracellular compartments, we observed an increase in temperature in the whole cell ($\Delta T = +1.4\,°C$), nucleus ($\Delta T = +0.9\,°C$), and cytoplasm ($\Delta T = +1.6\,°C$) (Fig. 5b, c). Intracellular temperature measurements using PEI-FNDs (FNDs surface-modified with polyethyleneimine [PEI]) also showed a similar differentiation-dependent temperature increase (Fig. 5d, e and Supplementary Note 3). By contrast, no temperature increase was observed when using unmodified FNDs positioned at the outer surface of the cells (Supplementary Fig. 9). Despite the different temperature

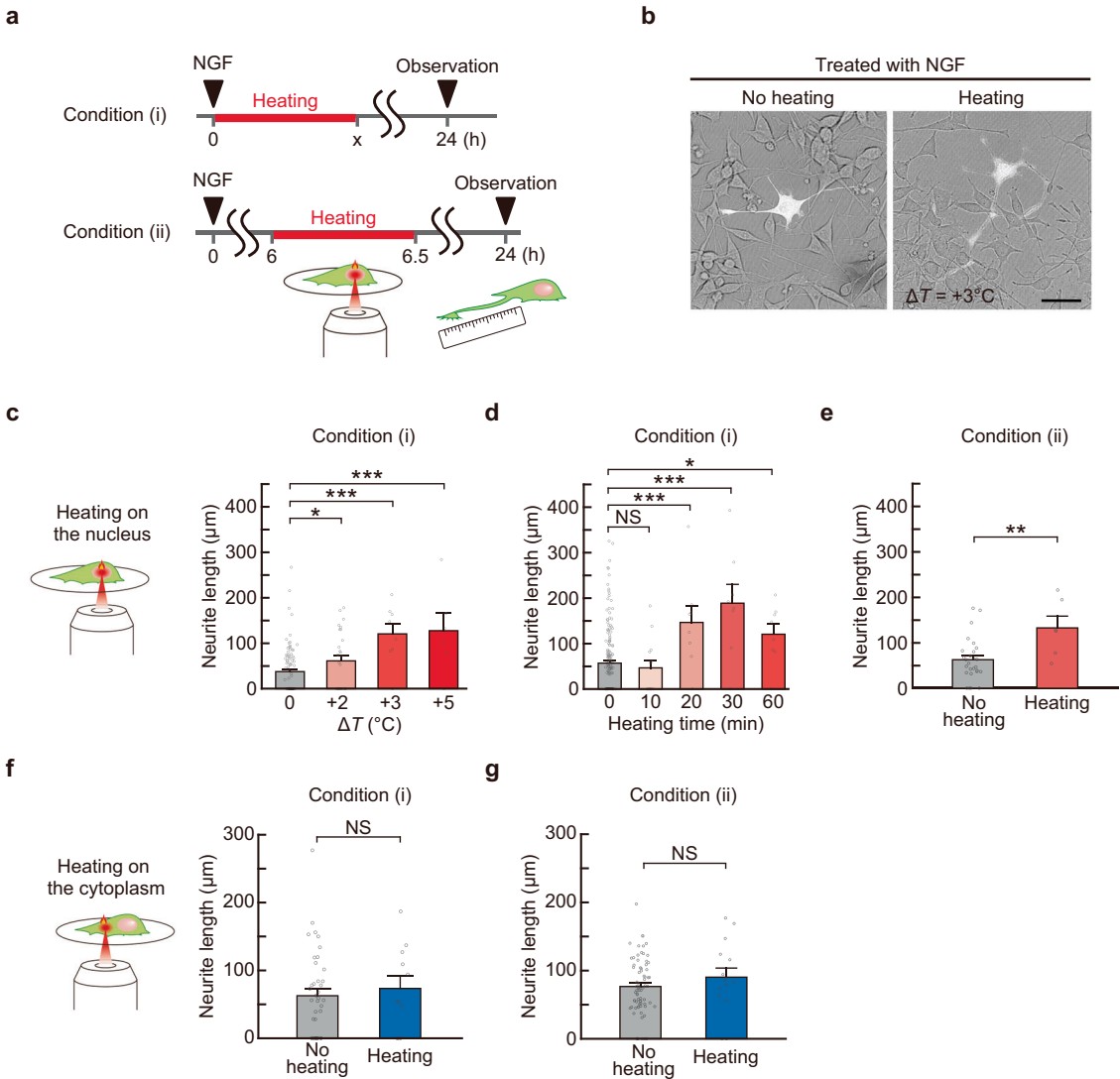

**Fig. 3 | Promotion of neurite outgrowth by local intracellular heating during neuronal differentiation of PC12 cells induced by NGF. a** Experimental schemes. Intracellular compartments were heated for a certain time after NGF treatment [condition (i)] or 30 min after 6 h of NGF treatment [condition (ii)], and neurite length was measured after 24 h of NGF treatment. **b** Representative pictures of cells after 24 h of NGF treatment. The bright field and fluorescence image of Texas Red-labeled dextran (cell-morphology marker) were merged. The scale bar represents 50 μm. **c** Influence of the intensity of IR laser heating (constant duration: 1 h) shown in (**b**) on neurite length after 24 h of NGF treatment. Data were obtained in condition (i) and are presented as means ± s.e.m. ($n$ = 96 [no heating], 27 [$\Delta T$ = +2 °C], 8 [+3 °C], and 6 [+5 °C] cells). **d** Influence of the duration of IR laser heating (constant intensity: $\Delta T$ = +3 °C) on neurite length after 24 h of NGF treatment in condition (i).

Data were presented as means ± s.e.m. ($n$ = 188 [no heating], 13 [heating: 10 min], and 8 [20, 30, and 60 min] cells). **e** Influence of nuclear heating ($\Delta T$ = +3 °C for 30 min) on neurite length after 24 h of NGF treatment in condition (ii). Data were presented as means ± s.e.m. ($n$ = 27 [no heating] and 6 [heating] cells). **f** Influence of cytoplasmic heating ($\Delta T$ = +3 °C for 30 min) on neurite length after 24 h of NGF treatment in condition (i). Data were presented as means ± s.e.m. ($n$ = 40 [no heating] and 11 [heating] cells). **g** Influence of cytoplasmic heating ($\Delta T$ = +3 °C for 30 min) on neurite length 24 h after NGF treatment in condition (ii). Data were presented as means ± s.e.m. ($n$ = 73 [no heating] and 15 [heating] cells). The number of independent experiments is the number of cells shown. $^*P < 0.05$, $^{**}P < 0.01$, $^{***}P < 0.001$ (one-sided unpaired Student's $t$-test). NS indicates not significant. Source data are provided as a Source Data file.

sensing principles of FPT and FND, both thermometers revealed that the process of neuronal differentiation was accompanied by spontaneous temperature increase (Supplementary Note 3).

### Involvement of intracellular reactions in the increase in intracellular temperature during neuronal differentiation
To investigate which intracellular reactions were involved in the temperature increase observed in PC12 cells during neuronal differentiation, we measured the intracellular temperature changes in the presence of inhibitors of the intracellular reactions associated with neuronal differentiation. Transcription, translation, and actin polymerization are the key intracellular reactions responsible for neuronal differentiation. When their reactions were inhibited using actinomycin

D (0.8 μM), cycloheximide (1.0 μM), and cytochalasin D (0.5 μM), respectively, neurite outgrowth was significantly reduced 24 h after the addition of inhibitors (Fig. 6a, b). The weak inhibitory effect of transcription inhibition by actinomycin D may be due to the fact that mRNAs with long half-lives remained in the cells. Intracellular temperature measurements using FPT$_{Low}$ showed that differentiated cells exhibited an increase in intracellular temperature in the absence of inhibitor treatment; there was no temperature increase in the nucleus and cytoplasm of cells in which the differentiation was inhibited by suppressing transcription, translation, and actin polymerization (Fig. 6c, d). Transcriptional inhibition significantly affected nuclear and cytoplasmic temperatures, whereas translational inhibition affected the cytoplasm more than nuclear temperature changes (Fig. 6d). Given

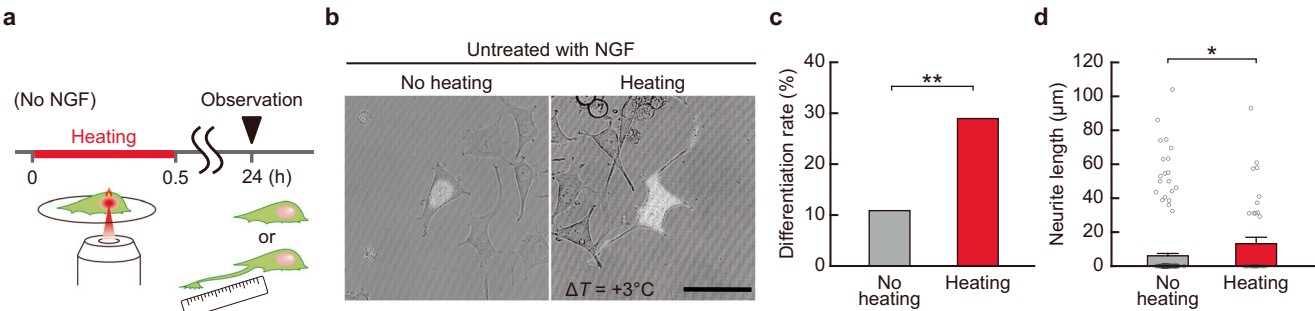

**Fig. 4 | Induction of neurite outgrowth in PC12 cells by local intracellular heating in the absence of NGF treatment. a** Time flow of local cellular heating experiments. The nucleus was heated for 30 min, and neurite length was measured after 24 h. **b** Representative pictures of cells 24 h after heating ($\Delta T = +3\,°C$ for 30 min) the nucleus. The bright field and fluorescence image of Texas Red-labeled dextran were merged. The scale bar represents 50 μm. **c** Neurite outgrowth rate 24 h after heating the nucleus in the absence of NGF treatment shown in (**b**). **d** Neurite length 24 h after heating the nucleus in the absence of NGF treatment shown in (**b**). Data were presented as means ± s.e.m. We analyzed 184 (no heating) and 38 (heating) cells. The number of independent experiments is the number of cells shown. $^{*}P < 0.05$, $^{**}P < 0.01$ (one-sided binomial test in **c** and one-sided unpaired Student's $t$-test in **d**). Source data are provided as a Source Data file.

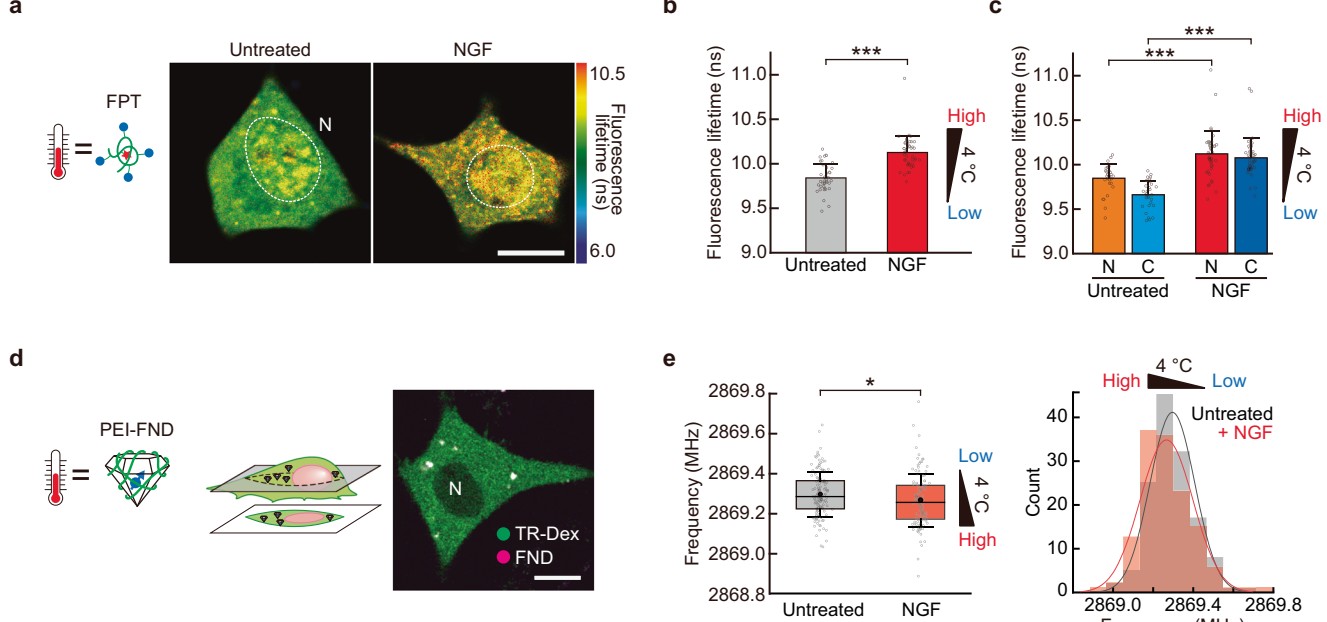

**Fig. 5 | Intracellular temperature increases during neuronal differentiation of PC12 cells. a** Intracellular temperature mapping before and after neuronal differentiation using FPT_Low. **b, c** Fluorescence lifetime of FPT_Low in whole cells (**b**), the nucleus (N), or the cytoplasm (C) (**c**) of untreated cells or cells treated with NGF shown in (**a**). Data were presented as means ± s.d. ($n = 40$ [untreated in **b**], 30 [untreated in **c**], and 44 [NGF] cells over three independent experiments). **d** Confocal fluorescence image of a cell having incorporated PEI-FNDs (right). Green and magenta are Texas Red-labeled dextran (TR-Dex, cell-morphology marker) and polyethyleneimine (PEI)-FNDs, respectively. **e** Intracellular temperature measurement during neuronal differentiation using ODMR of PEI-FNDs. Scattered plot (left; data were presented as box and whiskers plots showing the median, mean [black dots], interquartile range, s.d.) and histogram (right) of the resonance frequency of $D(T)$ of PEI-FND in untreated ($n = 136$ particles in 10 cells over three independent experiments) and NGF-treated cells ($n = 118$ particles in 12 cells over three independent experiments). The temperature of the medium was maintained at 30 °C. Scale bars represent 10 μm. $^{*}P < 0.05$, $^{***}P < 0.001$ (one-sided unpaired Student's $t$-test). N indicates the nucleus. Source data are provided as a Source Data file.

that transcription and translation occur in the nucleus and cytoplasm respectively, the decrease in temperature upon inhibition of transcription and translation indicates that transcription and translation (and their related reactions), rather than the function of specific gene products, are the source of heat. Furthermore, transcription and translation inhibition reduced the temperature below that at the untreated steady state, suggesting that the temperature of these compartments is maintained at a high level by thermogenesis even at a steady state. Although the inhibition of actin polymerization did not suppress the temperature increase above the NGF untreated level, it did prevent an NGF-dependent temperature increase, suggesting that actin polymerization and actin fiber-associated reactions during

neuronal differentiation were directly or indirectly involved in thermogenesis. These results demonstrated that intracellular transcription, translation, and actin polymerization reactions were involved in the increase in intracellular temperature during neuronal differentiation.

### Inhibition of neurite outgrowth through suppression of intracellular temperature increase during neuronal differentiation by NGF

We investigated whether a local increase in intracellular temperature was essential for neurite outgrowth during neuronal differentiation by analyzing the formation of NGF-induced protrusions when high

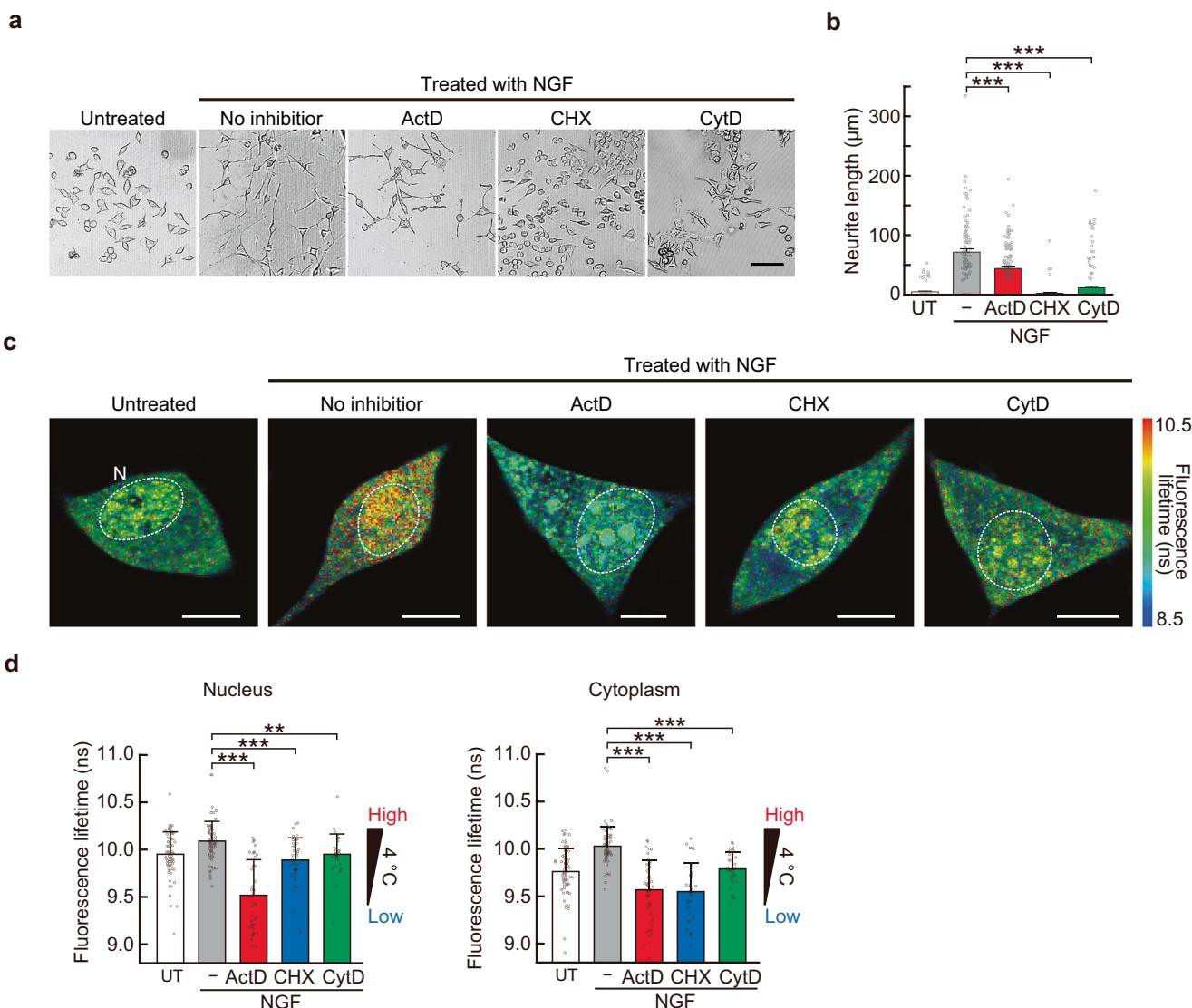

**Fig. 6 | Involvement of intracellular reactions in intracellular temperature increase during neuronal differentiation of PC12 cells. a** Bright-field images of cells after 24 h of NGF and inhibitor treatment. The scale bar represents 100 μm. **b** Length of neurites after 24 h of NGF and inhibitor treatment shown in (**a**). Data were presented as means ± s.e.m. ($n = 80$ [UT], 100 [NGF only], 120 [NGF, ActD], 97 [NGF, CHX], 180 [NGF, CytD] cells over two independent experiments). UT untreated, ActD actinomycin D, CHX cycloheximide, CytD cytochalasin D. **c** Intracellular temperature mapping after 24 h of NGF and inhibitor treatment.

Scale bars represent 10 μm. N indicates the nucleus. **d** The average fluorescence lifetime of $FPT_{Low}$ in cells after 24 h of NGF and inhibitor treatment is shown in (**c**). Data were presented as mean ± s.d. ($n = 70$ [UT], 71 [NGF only], 41 [NGF, ActD] for the nucleus, 45 [NGF, ActD] for the cytoplasm, 42 [NGF, CHX] for the nucleus, 37 [NGF, CHX] for the cytoplasm, and 31 [NGF, CytD] cells over three independent experiments). $^{**}P < 0.01$, $^{***}P < 0.001$ (one-sided unpaired Student's $t$-test). **c, d** The temperature of the medium was maintained at 30 °C. Source data are provided as a Source Data file.

concentrations of $FPT_{High}$ were introduced into the cells. The introduced $FPT_{High}$ absorbed large amounts of heat[48], which suppressed local intracellular temperature increase (Supplementary Fig. 10 and Supplementary Note 4). We examined the rate of differentiation (neurite outgrowth) and neurite length of cells cultured for 24 h after introduction of $FPT_{High}$ or control copolymer (CP: Supplementary Fig. 11 and Supplementary Note 4), which was performed immediately after the addition of NGF (Fig. 7a). CP-treated cells formed neurites upon addition of NGF (comparable to TR-Dex-treated cells), whereas neurite formation rate (Fig. 7b) and outgrowth (Fig. 7c) was markedly suppressed in cells injected with $FPT_{High}$. Additionally, IR laser irradiation of cells in which neurite outgrowth was inhibited by $FPT_{High}$ was followed by nuclear heating (150 μW for 30 min). This partially restored neurite outgrowth (Fig. 7d–f). We speculate that the partial recovery of neurite outgrowth by heating was caused by nonoptimal heating conditions (intensity, time, and location of heating) or partial

inhibition by $FPT_{High}$ due to factors other than heat. Our findings suggested that a local increase in cell temperature was necessary for neurite outgrowth during differentiation induced by NGF treatment.

## Contribution of thermal signaling to neurite outgrowth of primary mouse cortical neurons by measurement and manipulation of intracellular temperature

Finally, we investigated the physiological significance of the contribution of intracellular temperature increase to neurite outgrowth elucidated in PC12 cells using primary mouse cortical neurons. First, we measured the intracellular temperature of neurons using $FPT_{High}$ during neuronal circuit formation [days in vitro (DIV) 1, 4, 8: Fig. 8a]. Results showed that the intracellular temperature increased with the progression of days in culture during which the neurites elongated significantly (Fig. 8b, c). We then reduced this neuron-intrinsic local increase in intracellular temperature by introducing an excess amount

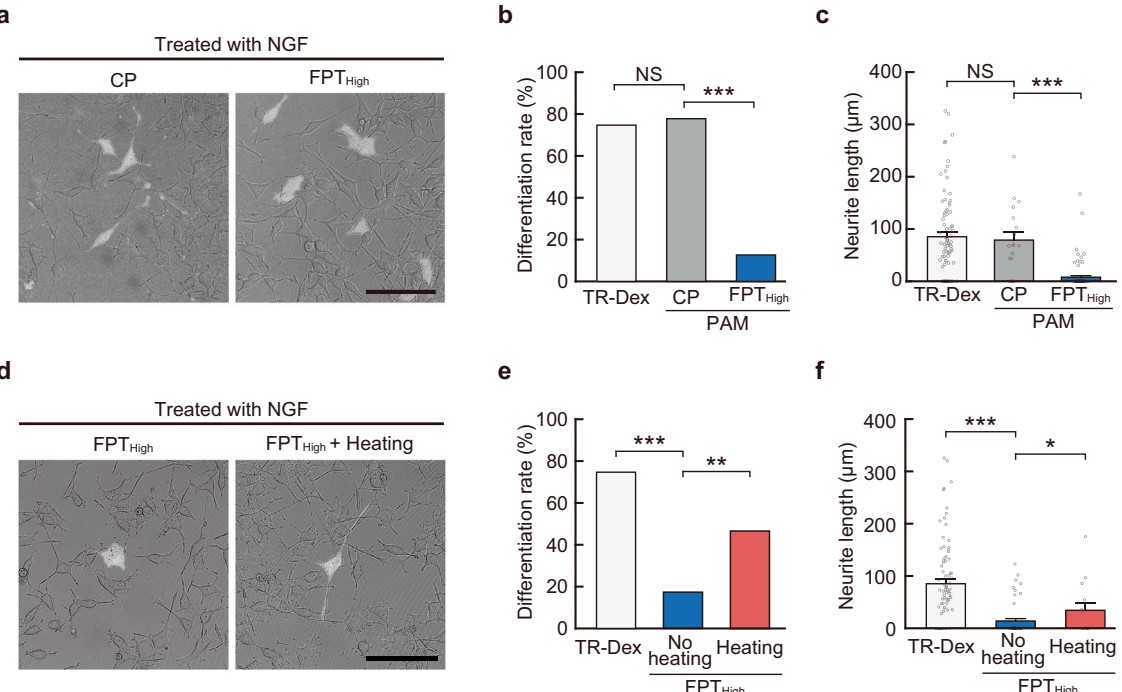

**Fig. 7 | Inhibition of neurite outgrowth by suppressing intracellular temperature increase during neuronal differentiation induced by NGF. a** Representative images of cells after 24 h of NGF treatment. The bright field and fluorescence images of cells containing control copolymer (CP, left) or FPT$_{High}$ (right) were merged. The scale bar represents 100 μm. **b, c** Outgrowth rate (**b**) and length (**c**) of neurites after 24 h of NGF treatment shown in (**a**). Data were presented as means ± s.e.m. For TR-Dex (cell tracer), 83 cells were analyzed. For polyacrylamides (PAMs), 18 (CP) and 87 (FPT$_{High}$) cells were analyzed over two independent experiments. **d** Representative images of cells after 24 h of NGF treatment and injection of FPT$_{High}$. The pictures were taken 24 h after heating ($\Delta T = +3$ °C for 30 min) the nucleus (right). The bright field and fluorescence images of FPT$_{High}$ in cells were merged. The scale bar represents 100 μm. **e, f** Outgrowth rate (**e**) and length (**f**) of neurites after 24 h of NGF treatment are shown in (**d**). Data were presented as means ± s.e.m. For TR-Dex, 83 cells were analyzed. For FPT$_{High}$, 52 (no heating) and 15 (heating the nucleus) cells were analyzed. The number of independent experiments is the number of cells shown. $^*P < 0.05$, $^{**}P < 0.01$, $^{***}P < 0.001$ (one-sided binomial test in **b**, **e** and one-sided unpaired Student's $t$-test in **c**, **f**). NS indicates not significant. Source data are provided as a Source Data file.

of FPT$_{High}$, which suppressed neurite outgrowth compared to neurons introduced with the same amount of CP. This suggests that a local increase in intracellular temperature is necessary for neurite outgrowth (Fig. 8d, e). These results are similar to those obtained with PC12 cells, suggesting that intracellular temperature increase contributes to neurite outgrowth of primary mouse cortical neurons.

## Discussion

In this study, we investigated the effects of spontaneous intracellular temperature changes on neuronal differentiation by locally manipulating and monitoring temperatures in PC12 cells and mouse primary cortical neurons. Our investigation of cellular responses to localized (<5 μm) heating by IR laser irradiation revealed that a certain duration (i.e., >20 min) and magnitude (i.e., 3 °C) of local temperature increase (especially in the nucleus) promoted or induced neurite outgrowth during neuronal differentiation. Furthermore, intracellular temperature measurements using FPTs and FNDs showed that the process of neuronal differentiation was accompanied by a spontaneous intracellular temperature increase. Additionally, reactions such as transcription in the nucleus and translation in the cytoplasm had a significant influence on heat generation during neuronal differentiation. Finally, the inhibition of local temperature increase by an excess dose of FPT$_{High}$ suggested that local heat generation within the cell played a key role in neurite outgrowth.

The originality of our approach to studying the mechanism of neuronal differentiation lies in the methods we developed that involve temperature manipulation. By applying an artificial and quantitative heating method that uses an IR laser which targets selectively subcellular compartments, we found that each compartment (i.e., nucleus

and cytoplasm) has a different temperature response that promotes protrusion elongation. Unlike previous studies that examined the effects of whole-cell temperature increases by manipulating the medium temperature[23–25] or external temperature gradient using extracellular heating[26,27], our approach, combined with the measurement of temperature in individual subcellular compartments by high-resolution temperature imaging, allowed us to verify the significance of temperature changes at subcellular sites. Recently, Choi et al. investigated the effects of intracellular heat on cell division in *C. elegans* embryos using a single-cell approach[42]. Our study shows that this elegant approach can be extended to intracellular compartments. Furthermore, a precise calibration of the heating intensity using a highly sensitive thermometer enabled a quantitative heating of ~1.3–8.0 °C in an area within a cell (which corresponds to ~0.2–1.5 °C when averaged over an entire cell). In particular, the heating conditions that most efficiently induced neurite outgrowth (3 °C locally and 0.5 °C for the whole cell) were noninvasive and similar to the magnitude of physiological temperature fluctuations. These findings have implications for investigating the functions of physiological heat generation in thermostatic organisms that are active in a narrow temperature range.

Our study revealed that noninvasive heating in the nucleus of PC12 cells promoted NGF-induced neurite outgrowth (Fig. 3) and that this nuclear heating alone also induced neurite outgrowth in absence of NGF (Fig. 4). We also found that not only did nuclear temperature spontaneously increase during neuronal differentiation in PC12 cells (Fig. 5), but that transcription, a biochemical reaction characteristic of the nucleus, was significantly involved in this temperature increase (Fig. 6). This intracellular thermogenesis caused by the stimulation of

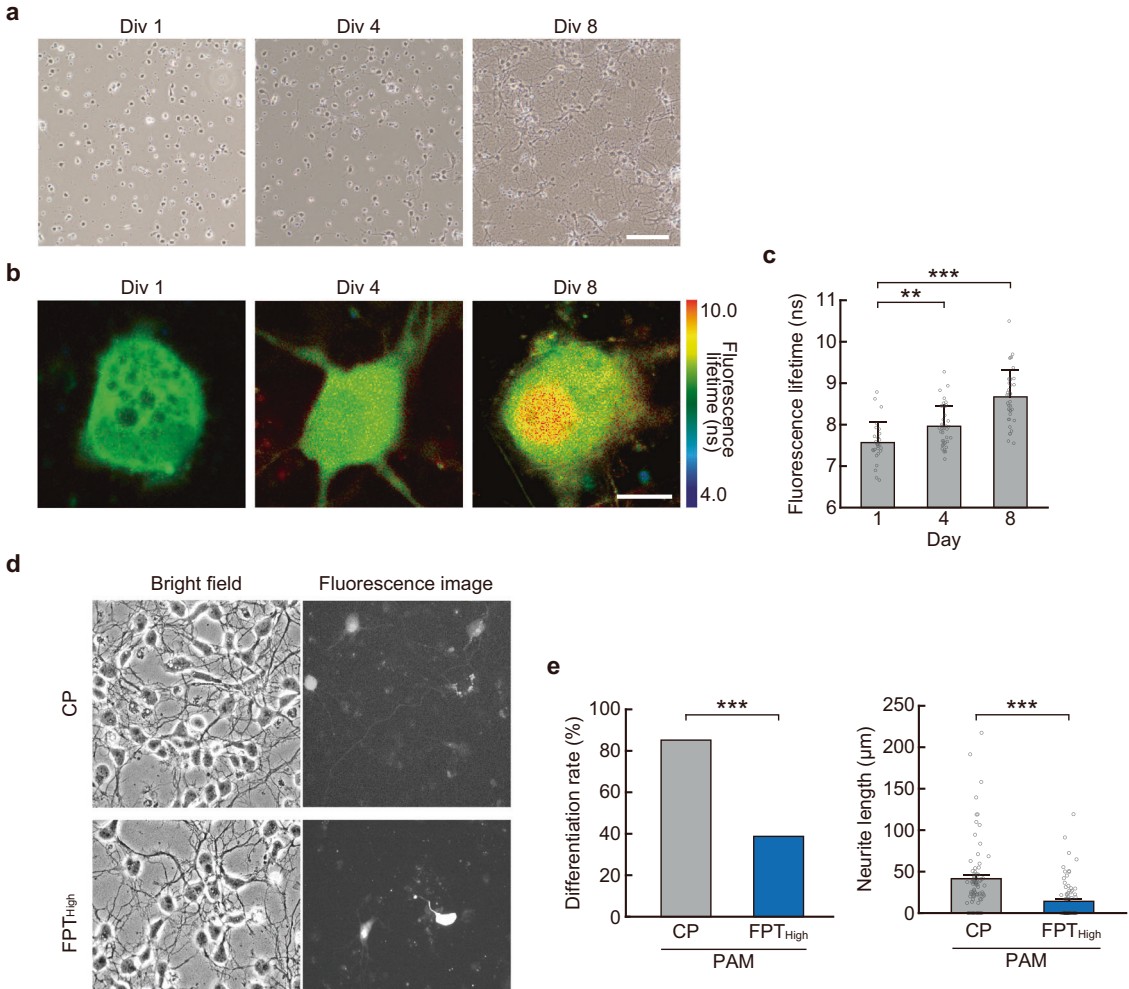

**Fig. 8 | Investigation of thermal signaling in neurite outgrowth of primary mouse cortical neurons by measurement and manipulation of intracellular temperature. a–c** Bright-field images (**a**), Representative intracellular temperature mapping (**b**), and the average fluorescence lifetime of FPT_High (**c**) of neurons at day in vitro (Div) are shown. Scale bars represent 100 μm (**a**) and 10 μm (**b**). The temperature of the medium was maintained at 37 °C. Data in (**c**) were presented as means ± s.d. (*n* = 28 [Day1], 38 [Day4], 37 [Day8] cells over two independent experiments). **d** Representative images of neurons 24 h after polymers (CP or FPT_High) introduction. The scale bar represents 50 μm. **e** Outgrowth rate and length of neurites 24 h after polymers introduction shown in (**d**). Data were presented as means ± s.e.m. For PAMs, 80 (CP over two independent experiments) and 81 (FPT_High over two independent experiments) cells were analyzed. $^{**}P < 0.01$, $^{***}P < 0.001$ (one-sided unpaired Student's *t*-test in **c**, **e** [right] and one-sided binomial test in **e** [left]). Source data are provided as a Source Data file.

intracellular reactions was necessary for neurite outgrowth (Fig. 7). In addition, an intracellular temperature increase also contributed to neurite outgrowth in primary mouse cortical neurons (Fig. 8). Based on these results, we offer a model in which physiological heat generation in the nucleus stimulates neurite outgrowth during neuronal differentiation through the positive regulation of transcription. In other words, we have uncovered the intracellular thermal signaling responsible for neuronal differentiation.

The response to local heating during neuronal differentiation is consistent with reports[23–25,29] showing that elevated environmental temperature promotes neuronal differentiation. Our study revealed that the nucleus is mainly involved in the response to physiological temperature increases and that heat is generated in a cell-autonomous manner. Considering that NGF activates numerous molecular cascades related to transcription[7,9], the present results suggest that heat promotes all or some specific reactions in these molecular cascades (Fig. 3), or acts as a signal in the cascade (Fig. 4). In other words, in addition to the effect of accelerating an already ongoing intracellular reaction, heat itself (via a change in the molecular and its complex state, etc.) is considered to be a trigger. Such a thermal signaling

mechanism might drastically activate transcription in response to stimuli. A thorough understanding of the molecular mechanism underlying heat-mediated facilitation of neuronal differentiation and potential positive feedback on transcription by generated physiological heat will require advances in intracellular local temperature manipulation techniques, such as heating of specific molecules and inhibition of localized temperature increases in the nucleus.

We also discovered that heat was generated in the cytoplasm during neuronal differentiation, resulting from translation reactions and actin polymerization (Fig. 6). Our results revealed that cytoplasmic heating slightly promoted neurite outgrowth (Fig. 3 and Supplementary Fig. 5), albeit not as efficiently as nuclear heating. This suggests translation activation by thermal signaling. To further investigate this activation, the development and optimization of effective heating methods (location, duration, and timing) in the cytoplasm, which is larger than the nucleus, are required.

Previous reports have discussed the effects of noxious extracellular temperature changes[23,24,28] (such as heat shock) and mild extracellular temperature stimuli[33] on neurite outgrowth. Here, we showed that cellular-intrinsic heat generation indirectly or directly

promoted neurite outgrowth. Our results suggest that what has been regarded as simply a by-product of biochemical reactions, heat generated spontaneously within the cell functions as a driving force for intracellular reactions. As such, we hypothesize that thermal signaling can result in highly efficient neurite outgrowth even under physiological conditions. For example, spontaneous increases in body temperature during a critical period of organogenesis (2–8 weeks after fertilization)[49] may promote the formation of neural circuits in the fetus. In addition, the related findings of our study, such as the intracellular location (nucleus) and timing of heating required to induce neurite outgrowth, may contribute to developing rehabilitation and neuroregenerative therapies for which neurogenesis is essential.

Uncovering the detailed molecular mechanisms of neuronal differentiation mediated by intracellular temperature changes will deepen our understanding of the mechanisms and significance of complex neuronal differentiation. Our discovery that neuronal differentiation is driven by spontaneous intracellular temperature changes suggests that this dynamic and complex cellular phenomenon is indeed one of the unique mechanisms of thermal signaling[41] that has been proposed, but not confirmed, by any previous studies.

## Methods

### Study approval
For animal studies, all the procedures were performed in accordance with the animal health care guidelines of Japan and approved by the ethics committee of AIST (permission No. 2023-0008).

### Cell culture
Neuron-like PC12 cells were purchased from Riken Bioresource Center (No. RCB0009, Tsukuba, Japan), cultured in Dulbecco's modified Eagle's medium (D6429, Sigma-Aldrich, St. Louis, MO) supplemented with 5% thermo-inactivated horse serum (Gibco), 5% fetal bovine serum (Biowest, Nuaillé, France), 50 mg/ml streptomycin, and 50 IU/ml penicillin (15140122, Thermo Fisher, Waltham, MA), and maintained in a humidified atmosphere of 95% air/5% $CO_2$ at 37 °C. For microscopic observation, cells were seeded onto poly-L-lysine-coated glass bottom dishes (D11131H MATSUNAMI, Kishiwada, Japan) at a density of $5 \times 10^4$ cells/dish.

### Preparation of $FPT_{High}$
NiPAM (2.4 mmol), 3-(acrylamidopropyl)trimethylammonium (APTMA, 0.1 mmol, Aldrich), DBThD-AA (25 µmol), and α,α′-azobisisobutyronitrile (25 µmol) were dissolved in $N,N$-dimethylformamide (5 ml), and the solution was bubbled with dry argon for 30 min to remove dissolved oxygen. The solution was heated to 60 °C for 15 h and then cooled to room temperature. The reaction mixture was then poured into diethyl ether (200 ml). The obtained $FPT_{High}$ was purified by dialysis (yield: 45%).

The contents in NiPAM and APTMA units of the copolymers were determined from 1H-nuclear magnetic resonance spectra (Bruker AVANCE400). The proportions of DBD-AA units in the copolymers were determined from their absorbance in methanol compared with the model fluorophore $N$-,2-dimethyl-$N$-(2-{methyl[7-(dimethylsulfamoyl)-2,1,3-benzoxadiazol-4-yl]amino}ethyl)propanamide (DBD-IA) ($\varepsilon = 10{,}800\,M^{-1}\,cm^{-1}$ at 444 nm). The molecular weight of $FPT_{High}$ was determined using gel permeation chromatography with weight average molecular weight (Mw) = 17,108 and number average molecular weight (Mn) = 7788. A calibration curve was obtained using a polystyrene standard, and 1-methyl-2-pyrrolidinone containing LiBr (5 mM) was used as eluent.

### Microinjections into cells
$FPT_{Low}$, which was described in the previous report[34], and TR-Dex (D1828, Thermo Fisher), which was used as a cell tracer, were dissolved in an aqueous solution (0.5% w/v) containing 130 mM KCl, 10 mM

$K_2HPO_4$, and 22 mM NaCl. The solution was filtered using an Ultrafree-MC filter (Millipore, Burlington, MA) and microinjected into the cytoplasm using a glass capillary needle (Femtotips II, Eppendorf, Hamburg, Germany). The volume of the injected solution was estimated to be 2 fl.

### Induction and analysis of neurite outgrowth
To induce neural differentiation, NGF (N-100, Alomone Labs, Jerusalem, Israel) dissolved in phosphate-buffered saline (27575-31, Nacalai Tesque, Kyoto, Japan) containing 0.1% w/v bovine serum albumin (A-6003, Sigma-Aldrich) was added to the culture medium lacking serum at a final concentration of 50 ng/ml. Intracellular temperature changes and neurite outgrowth associated with differentiation were measured 24 h after NGF addition during differentiation (Fig. 1b) unless otherwise indicated. Protrusions were observed by bright-field imaging using a TCS SP8 confocal laser-scanning microscope (Leica Microsystems, Wetzlar, Germany). To track thermally manipulated cells over time, the shape of the cells, including the protrusions, was visualized using TR-Dex and observed under a confocal fluorescent microscope. Neurite length was defined as the length of the longest protrusion that was longer than the cell body. Therefore, protrusions shorter than the cell body were not considered as neurites, and were assigned a value of 0 µm for statistical analyses. For the measurement of neural differentiation rate, cells with protrusions longer than the cell body were considered differentiated cells. Cells that detached from the dish or displayed significant morphological alterations following microinjection, laser heating, or treatment with inhibitors were excluded from the analysis.

### Neurite imaging and FPT-based temperature imaging of the cells
Confocal fluorescence imaging and fluorescence lifetime imaging of PC12 cells were performed using 35-mm poly-L-lysine-coated glass bottom dishes (D11131H MATSUNAMI) and a confocal laser-scanning microscope (Leica Microsystems). The fluorescence lifetime of FPTs was measured using a TCSPC system (PicoHarp 300, PicoQuant, Berlin, Germany). PC12 cells were excited with a pulsed diode laser (PDL800-B, 01029349, 470 nm, PicoQuant) at a repetition rate of 20 MHz, and the emission from 500 to 700 nm was collected through an HC PL APO 63 × 1.4 oil CS2 objective (Leica Microsystems) in a 128 × 128 pixel format. For fluorescence lifetime imaging in localized intracellular regions during IR laser heating (Fig. 2), a highly sensitive TCSPC system Falcon (Leica Microsystems) was used. To visualize neurite outgrowth, TR-Dex in PC12 cells was excited with a 552-nm laser, and the emission from 560 to 700 nm was collected through an HC PL APO 40 × 1.3 oil CS2 objective (Leica Microsystems) in a 1024 × 1024 pixel format. The cell medium temperature was controlled using a stage-top incubator (SPG-WSKMX-SET, Tokai Hit, Fujinomiya, Japan). Fluorescence lifetime images were acquired by accumulating 10–20 times. The obtained fluorescence decay curve for a given region was fitted with a double exponential function using Symphotime64 software (PicoQuant) for the PicoHarp 300 system or LAS-X software (Leica Microsystems) for the Falcon system.

The average fluorescence lifetime calculated as previously reported[34] was used as a temperature-dependent variable. To calibrate the intracellular temperature from the responses of FPTs, a linear approximation of the relationship between the averaged fluorescence lifetime of FPTs in a PC12 cell and the temperature of the medium was made (Fig. 1):

$$\text{for } FPT_{Low} \qquad \tau_f(T) = 0.2046T + 3.505 \qquad (1)$$

$$\text{for } FPT_{High} \qquad \tau_f(T) = 0.3005T + 2.724 \qquad (2)$$

where $T$ and $\tau_f(T)$ represent the temperature (°C) and the fluorescence lifetime (ns) at $T$°C, respectively. Although a method of approximating the calibration curve of FPT by a polynomial has been established[34], in this study, the FPT response was linearly approximated because it was calibrated from the proportional response over narrow temperature ranges. Cell compartments with a maximum number of photons per pixel of less than 1000, regions discerned by an averaged analysis, were not subjected to fluorescence lifetime determination.

## Temperature measurements in cells using FNDs

We prepared FNDs with positively charged surfaces by modifying them with PEI to accommodate the negative charges of the cell membrane and, consequently, increase the efficiency at which FNDs entered cells. A 0.1% w/v PEI solution was added to unmodified FNDs, the mixture was vortexed, sonicated, and stirred for 1 h. Then, FNDs were collected by centrifugation ($10,394 \times g$, 3 min) and washed twice with Milli-Q water. The size and surface charge of the FNDs were measured using a Zetasizer Nano ZSP (Malvern Panalytical, Malvern, UK) (Supplementary Fig. 12). The obtained PEI-FNDs and unmodified FNDs were mixed in serum-free medium to final concentrations of 5 and 10 µg/ml, respectively, and incubated with PC12 cells at 37 °C, 5% $CO_2$ for 3 and 4 h, respectively, for uptake into the cells.

Temperature measurements in cells using FNDs were performed with the home-built microscope, which was described previously[50]. A continuous Nd: YAG laser (532 nm) illuminated the FNDs to initialize and read out the spin state of the nitrogen-vacancy centers (NVCs) on an inverted microscope system (Ti–E, Nikon, Tokyo, Japan). ODMR spectra were recorded for a range of microwave frequencies as the difference between the fluorescence intensities with and without microwave irradiation. The temperature of the stage was set at 33 °C using a stage-top incubator (TPi-108RH26, TOKAI HIT) on an electric moving stage.

Fluorescence images were recorded while the microwave frequency was digitally swept across a resonant frequency range of 2850 to 2890 MHz in increments of 0.4 MHz. Through this process, ODMR spectra were obtained from single FND particles in PC12 cells. The exposure time of the camera was 5 ms. The accumulation of the fluorescence intensities of the two images for each microwave frequency was repeated 32 times. Then, each ODMR spectrum was fitted to the sum of the two Lorentzian functions to determine the zero-filed splitting D. D was defined as the central value of the two resonances for ms = 0 to 1 and ms = 0 to −1. The measured temperature change was determined based on the measured thermal shift in the ODMR peak, $\Delta D/\Delta T = -0.068$ MHz/°C (Fig. 1g), and we estimated the temperature change of the FNDs before and after the differentiation of PC12 cells.

## Local intracellular heating using IR irradiation

Local intracellular heating was conducted using an IR laser (1475 nm, Sigma-Koki, Tokyo, Japan) installed on a TCS SP8 confocal laser-scanning microscope (Leica Microsystems). The IR laser was irradiated through an HC PL APO 63 × 1.4 oil CS2 objective (Leica Microsystems) controlled by a mechanical shutter (SSH-C2B, Sigma-Koki). The temperature of the medium during heating was set at 37 °C ± 0.1 °C with a stage-top incubator (STG-WSKMX-SET, TOKAI HIT). To calibrate the intracellular temperature increase ($\Delta T$) from the laser intensity-dependent FPT_High response, we calculated the temperature change from the change in FPT fluorescence lifetime in the IR laser-irradiated area in PC12 cells based on a linear approximation of the linear relationship between FPT fluorescence lifetime and laser power.

$$\text{for heating in the nucleus} \quad \tau_f = 0.01025P - 0.5995 \quad (3)$$

$$\text{for heating in the cytoplasm} \quad \tau_f = 0.01069P - 0.7793 \quad (4)$$

where $\tau_f$ and $P$ represent the fluorescence lifetime (ns) and the power of IR-laser power (µW), respectively.

## Inhibition of intracellular reactions

To investigate the involvement of intracellular reactions on intracellular temperature during neuronal differentiation, 0.8 µM actinomycin D (595-00261, FUJIFILM Wako Pure Chemical Corporation, Osaka, Japan), 1.0 µM cycloheximide (C7698, Sigma-Aldrich), and 0.5 µM cytochalasin D (037-17561, FUJIFILM Wako Pure Chemical Corporation) were used as inhibitors of transcription, translation, and actin polymerization, respectively, during neuronal differentiation. The compounds were dissolved in dimethyl sulfoxide (DMSO) mixed with a medium at their respective concentrations and added to the cells by a medium exchange. The final DMSO concentration in the medium was 0.05–0.1% and had no effect on neurite outgrowth in PC12 cells (Supplementary Fig. 13).

## Preparation and observation of primary mouse cortical neurons

Primary cortical neurons were prepared from embryonic 14–15-day fetal mice (C57BL/6J, male and female, wild type) (CLEA Japan, Tokyo, Japan). Cortical tissues dissected from the brain were treated with trypsin (0.25%, Nacalai Tesque, Kyoto, Japan) for 20 min in Hank's balanced solution, and neurons were gently dissociated. The obtained neurons were plated on poly-ethylenimine (0.1%, Sigma) coated glass bottom dishes at a density of $2–5 \times 10^4$ cells/dish. Neurons were cultured in MEM culture medium (Nissui, Tokyo, Japan), supplemented with L-alanyl-L-glutamine (0.5 mM, Nacalai Tesque), fetal bovine serum (2%), NeuroBrew (2%, Miltenyi Biotec, Tokyo, Japan), HEPES (10 mM, pH7.4), and glucose (4.5 g/L).

To investigate the thermal effect on neurite extension in neurons, the polymers (FPT_High and control copolymer) were introduced into neurons with a solution of iso-osmotic glucose solution (5%) for 5 min at 4 °C. The shape of the cells, including the protrusions was visualized by the fluorescence of the loaded polymers, and the length of the neurites was measured using Fiji software.

## Immunofluorescence imaging of MAP2

PC12 cells were fixed with 4% paraformaldehyde in PBS (4% sucrose) for 5 min at room temperature, treated with 50 mM $NH_4Cl$ in PBS for 5 min, permeabilized with 0.1% Triton X-100 (Wako) in PBS for 5 min at room temperature and blocked with Blocking One (Nacalai Tesque) after washing with PBS. Subsequently, the cells were incubated with a mouse monoclonal anti-MAP2 antibody (dilution 1:500) (M4403, Sigma-Aldrich). The primary antibodies were detected with secondary anti-mouse IgG (H + L) antibodies labeled with Alexa Fluor 647 (dilution 1:1000) (715-605-150, Jackson ImmunoResearch Laboratories, PA, USA). After being washed three times with PBS, the cells were imaged using a TCS SP8 confocal laser-scanning microscope (Leica Microsystems).

## Statistical analysis and reproducibility

Statistical significance was determined by one-sided unpaired Student's $t$-test and one-sided binomial test using Excel software (Microsoft, Redmond, WA). All representative micrographs were reproduced the number of times indicated in the accompanying plot analyses. No statistical method was used to predetermine the sample size. Cells that detached from the dish or displayed significant morphological alterations following microinjection, laser heating, or treatment with inhibitors were excluded from the analysis.

## Reporting summary

Further information on research design is available in the Nature Portfolio Reporting Summary linked to this article.

## Data availability

All data that support the findings of this study are included in the paper. Microscopy data used for analyses are available from the corresponding author upon request. Requests will be fulfilled within 1 month. The data generated in this study are provided in the Source Data file. Source data are provided with this paper.

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

## Acknowledgements

We are grateful for support from JSPS KAKENHI Grant Numbers 15H05931 (K.O. and Y.H.), 17H03075 (S.U.), 20H05785 (K.O. and Y.H.), 22H02583 (Y.H.), 23KJ1450 (S.C.) and 24H02306 (K.O. and Y.H.), MEXT Q-LEAP Grant Number JPMXS0120330644 (Y.H.), Mr. Hirotaka Okita and Ms. Eiko Fukatsu for technical support.

## Author contributions

K.O. and Y.H. designed this work. S.C., T.A., M.K., K.O. and Y.S. performed all measurements and manipulations of intracellular temperature using FPTs and IR laser in PC12 cells. S.C. and S.S. prepared FND sensors and performed FND-based thermometry in single cells. S.U. prepared FPT$_{High}$. K.K. and S.C. performed experiments using primary mouse cortical neurons. S.C. analyzed the data. S.C., K.O. and Y.H. wrote the paper. K.O. organized this work. All authors discussed the results and commented on the manuscript.

## Competing interests

The authors declare no competing interests.
