## [Peer Review File · Nature Communications]

Reviewers' Comments:

Reviewer #1:

Remarks to the Author:

The study from Chuma et al., examines the role of thermal signalling in a simple in vitro model of neuronal differentiation. The experiments are extremely well executed. The authors demonstrate that upon neuronal differentiation the intracellular temperature increases. Neuronal differentiation can be accelerated by increasing the temperature within the nucleus. On the other hand, preventing temperature increase through clever use of a molecule known as FPT can slow the rate of neural differentiation.

In general, this paper is technically sound, well presented and well-written, and I did find it of interest. However, I do not believe the paper will be of broad interest to the readership of Nature Communications. The fact that increasing temperature would accelerate a biophysical process that is already underway (namely neuronal differentiation) seems to me more akin to a restatement of a fact of physics & chemistry, rather than revealing new biological mechanisms. I appreciate this study has a number of points of technical advancement (e.g., finer manipulation of temperature within a cell), however, the significance of these findings seems too minor. The study also entirely occurs in vitro, within a single cell line, trying to tie this phenomena to an in vivo process or in vivo manipulation might increase enthusiasm for the novelty of the study.

Technical issue:

1) Ln 217-219: In the discussion, it is mentioned that the heat increase is 'necessary' for neurite outgrowth. The word 'necessary' should not be used as neurite outgrowth still occurs just at a slower rate

Reviewer #2:

Remarks to the Author:

The manuscript was well written in general. The work is of significance to the related fields. The work supports the conclusions and claims of the manuscript. The methodology is sound enough. There is enough detail provided in the methods for the work to be reproduced.

It can be accepted after a minor revision. My suggestions on the manuscript are as follows:

The sentence "Intracellular thermometry revealed that neuronal differentiation was accompanied by intracellular thermogenesis associated with transcription and translation and that this thermogenesis was required for differentiation." in the Abstract is too complicated to understand.

The relative standard deviation (RSD) shown in Figure 2 is large. Do authors explain the reason?

The title of references should be lowercase.

Two spaces are needed in the start of each paragraph.

The English of the manuscript need be improved.

Reviewer #3:

Remarks to the Author:

In this manuscript, the authors investigate specific intracellular and nuclear temperature variations during NGF-induced neuronal differentiation and observe that spontaneous temperature changes occur associated with transcription and translation.

The authors applied innovative approaches for both thermal stimulation and determination of localized intracellular heating. The approaches are interesting because enable stimulation and analysis localized in a specific region of the single cell.

However, there are some issues that should be addressed which would improve the manuscript.

1) Concerning the results of the preliminary experiments of localized intracellular heating, the authors write in the manuscript that high temperature differences ($\pm 5^{\circ}\text{C}$) induced cytotoxicity, which has been described as an increase in cell detachment from the dish during or after heating

(lines 127-129). For instance, it would be worth showing results of viability/toxicity for the different temperature increases (+2, +3 and +5°C) to support that temperature increase of +5°C is cytotoxic, whereas $\Delta T = +2^\circ\text{C}$ and $+3^\circ\text{C}$ does not induce harmful effects and represents a more physiological condition.

2) The acquisition of a differentiated phenotype is not sufficiently supported, since the authors investigate only neurite outgrowth and the number of cells with protrusions longer than their cell bodies. The authors should at least confirm neuronal differentiation by the analysis of the expression of neuronal markers, such as for example NeuN, which is a standard marker for mature neurons.

3) Referring to Figure 6, the authors write that neurite length decreases because of the administration of the different inhibitory agents. However, it seems that cells are suffering, and I wonder if effects on neurite retraction could be due to a change in cell morphology related to the stress and maybe to high inhibitor concentrations. I would ask the authors to elucidate this issue.

4) In the manuscript, lines 179-180, the authors write: "the temperature decrease was dependent on the intracellular compartments as inhibition of transcription affected nuclear and cytoplasmic temperatures, whereas inhibition of translation affected only the cytoplasm (Figure. 6d). However, in Figure 6D all the temperature are significantly decreased. Could the authors explain their assumption which is in contrast with the data they show?"

5) The employment of FNDs should be better explained, since the authors sometimes write that the use of FNDs is for extracellular temperature measurement, whereas in some other parts of the manuscript they write about intracellular temperature (lines 513-514).

6) Please, add statistical analysis for data in Figure 1b.

Some minor revisions and typo corrections are also required through the manuscript. For example:

In line 44, "Not only is brain function," please correct "is" with "in".

Please, reformulate the sentence in lines 170-171.

Please, correct line 517: $ms = 0, \pm 1$.

[Point-by-point responses and revisions in accordance with reviewer's comments]

Ms ID: NCOMMS-23-25324

Ms title: Implication of thermal signaling in neuronal differentiation revealed by manipulation and measurement of intracellular temperature

Reviewer #1 (Remarks to the Author)

Comment: The study from Chuma et al., examines the role of thermal signalling in a simple in vitro model of neuronal differentiation. The experiments are extremely well executed. The authors demonstrate that upon neuronal differentiation the intracellular temperature increases. Neuronal differentiation can be accelerated by increasing the temperature within the nucleus. On the other hand, preventing temperature increase through clever use of a molecule known as FPT can slow the rate of neural differentiation.

Response: We appreciate that Reviewer #1 properly summarized and valued our experiments.

Comment: In general, this paper is technically sound, well presented and well-written, and I did find it of interest. However, I do not believe the paper will be of broad interest to the readership of Nature Communications. The fact that increasing temperature would accelerate a biophysical process that is already underway (namely neuronal differentiation) seems to more akin to a restatement of a fact of physics & chemistry, rather than revealing new biological mechanisms. I appreciate this study has a number of points of technical advancement (e.g., finer manipulation of temperature within a cell), however, the significance of these findings seems too minor.

Response: The Arrhenius equation indicates that the rate of biochemical reactions increases with increasing temperature. In contrast, in this study, the results of the neuronal differentiation of PC12 cells by heating alone (Figure 4), despite the absence of NGF, suggest that not only does the reaction rate of a process already underway increase, but that a completely different mechanism also contributes. In other words, the increase in temperature is considered here to act as a trigger for the reaction, e.g., via a change in the molecular and its complex state, etc. Therefore, it is suggested that there are two mechanisms of thermal signaling described in this paper: acceleration and the triggering of intracellular reactions. Furthermore, the temperature changes detailed in this study were dependent on intrinsic heat sources rather than externally applied environmental temperature changes (Figures 5, 6, 8). These results suggest that heat, previously considered to be a by-product of biochemical reactions in cell biology, when generated spontaneously within the cell serves as a driving force for intracellular reactions. This may provide a new physiological role for the previously known cellular heat response system and the temperature heterogeneity of each organelle. These two mechanisms are novel concepts in cell biology, not to mention neuronal differentiation, and the implications of this study are significant. Therefore, to clarify this point, the following sentences have been added in

the revised manuscript:

Text added to the revised manuscript (Results and Discussion sections):

“These effects of heating in the absence of NGF suggest that the local temperature increase in the nucleus itself functions as a trigger for neuronal differentiation. This may be activated via a change at the molecular level and in its complex state (etc.) rather than via an acceleration of an already ongoing intracellular reaction due to an increase in temperature.” (Line 163-166 in the revised manuscript)

“In other words, in addition to the effect of accelerating an already ongoing intracellular reaction, heat itself (via a change in the molecular and its complex state, etc.) is considered to be a trigger.” (Line 280-281 in the revised manuscript)

“Previous reports have discussed the effects of noxious extracellular temperature changes^{23,24,28} (such as heat shock) and mild extracellular temperature stimuli³³ on neurite outgrowth. Here, we showed that cellular-intrinsic heat generation indirectly or directly promoted neurite outgrowth. Our results suggest that what has been regarded as simply a by-product of biochemical reactions, heat generated spontaneously within the cell functions as a driving force for intracellular reactions.” (Line 293-297 in the revised manuscript)

Comment: The study also entirely occurs *in vitro*, within a single cell line, trying to tie this phenomena to an *in vivo* process or *in vivo* manipulation might increase enthusiasm for the novelty of the study.

Response: As suggested by Reviewer #1, to test the physiological significance of the findings of this study, we performed additional experiments related to the *in vivo* process. We performed intracellular temperature measurements and local intracellular temperature manipulations during neurite outgrowth of primary cortical neurons derived from fetal mice. These results indicate that thermal signaling drives neurite outgrowth of primary mouse cortical neurons. This result strongly supports that the contribution of thermal signaling observed in PC12 cells occurs *in vivo* and thus significantly increases the novelty of this study. Therefore, we have added the following to this paper:

Text and figure added to the revised manuscript (Results and Discussion sections):

“Contribution of thermal signaling to neurite outgrowth of primary mouse cortical neurons by measurement and manipulation of intracellular temperature.

Finally, we investigated whether the contribution of thermal signaling to neurite outgrowth elucidated in PC12 cells also occurs *in vivo* using primary mouse cortical neurons. First, we measured the intracellular temperature of neurons using FPT_{High} during neurite outgrowth [days *in vitro* (DIV) 1, 4, 8: Figure 8a]. The results showed that the intracellular temperature increased with the progression of days in culture, during which the neurites

elongated significantly (Figure 8b,c). We then reduced this neuron-intrinsic local increase in intracellular temperature by introducing an excess amount of FPT_{High}, which suppressed neurite outgrowth compared to neurons introduced with the same amount of control copolymer (CP). This suggests that a local increase in intracellular temperature is necessary for neurite outgrowth (Figure 8d,e). These results are similar to those obtained with PC12 cells, suggesting that intracellular temperature increase contributes to neurite outgrowth of primary mouse cortical neurons.” (Line 226-236 in the revised manuscript)

“In addition, intracellular temperature increase also contributed to neurite outgrowth in primary mouse cortical neurons (Figure 8).” (Line 270-271 in the revised manuscript)

Figure 8. Investigation of thermal signaling in neurite outgrowth of primary mouse cortical neurons by measurement and manipulation of intracellular temperature. (a-c) Bright field images (a), Representative intracellular temperature mapping (b), and the average fluorescence lifetime of FPT_{High} (c) of neurons at day *in vitro* (Div) shown. Scale bars represent 100 μm (a) and 10 μm (b). (d) Representative images of neurons 24 h after polymers (CP or FPT_{High}) introduction. Scale bar represents 50 μm. (e) Outgrowth rate and length of

neurites 24 h after polymers introduction. Data are presented as means \pm s.e. For PAMs, 80 (CP) and 81 (FPT_{High}) cells were analyzed.

Technical issue:

Comment: 1) Ln 217-219: In the discussion, it is mentioned that the heat increase is ‘necessary’ for neurite outgrowth. The word ‘necessary’ should not be used as neurite outgrowth still occurs just at a slower rate

Response: As Reviewer #1 pointed out, we have revised the relevant text in the discussion

Text added to the revised manuscript (Discussion section): “Finally, the inhibition of local temperature increase by an excess dose of FPT_{High} suggested that local heat generation within the cell played a key role in neurite outgrowth.” (Line 248-249 in the revised manuscript)

Reviewer #2 (Remarks to the Author)

Comment: The manuscript was well written in general. The work is of significance to the related fields. The work supports the conclusions and claims of the manuscript. The methodology is sound enough. There is enough detail provided in the methods for the work to be reproduced.

It can be accepted after a minor revision. My suggestions on the manuscript are as follows:

Response: We deeply appreciate Reviewer #2’s positive comments on our manuscript and constructive ideas for the revision.

Comment: The sentence “Intracellular thermometry revealed that neuronal differentiation was accompanied by intracellular thermogenesis associated with transcription and translation and that this thermogenesis was required for differentiation.” in the Abstract is too complicated to understand.

Response: As Reviewer #2 pointed out, we have revised the relevant text in the abstract:

Text added to the revised manuscript (Abstract): “Intracellular thermometry revealed that neuronal differentiation was accompanied by intracellular thermogenesis associated with transcription and translation. Suppression of intracellular temperature increase during neuronal differentiation inhibited neurite outgrowth.” (Line 35-39 in the revised manuscript)

Comment: The relative standard deviation (RSD) shown in Figure 2 is large. Do authors explain the reason?

Response: We speculate that the variation in temperature change (ΔT) may be due to the potential differences in thermodynamic environments between cells or between local regions within cells (i.e., cells or regions that are easily warmed and regions that are not). Since the heated region in this study is small (5 μm), these local environmental effects are unlikely to be averaged out. Therefore, we have added these reasons in the revised manuscript:

Text added to the revised manuscript (Results section): “We hypothesize that the variation in temperature change (ΔT) may be due to the potential differences in thermodynamic environments between cells or between local regions within cells (i.e., cells or regions that are easily warmed and regions that are not). As the heated region in this study is small (5 μm), these local environmental effects are unlikely to be averaged out.” (Line 119-123 in the revised manuscript)

Comment: The title of references should be lowercase.

Response: As Reviewer #2 pointed out, we revised the format of the references.

Comment: Two spaces are needed in the start of each paragraph.

Response: As Reviewer #2 pointed out, we revised the format of the manuscript.

Comment: The English of the manuscript need be improved.

Response: The manuscript was revised throughout with regards to English grammar and readability.

Reviewer #3 (Remarks to the Author)

Comment: In this manuscript, the authors investigate specific intracellular and nuclear temperature variations during NGF-induced neuronal differentiation and observe that spontaneous temperature changes occur associated with transcription and translation.

The authors applied innovative approaches for both thermal stimulation and determination of localized intracellular heating. The approaches are interesting because enable stimulation and analysis localized in a specific region of the single cell.

Response: We appreciate that Reviewer #3 properly summarized and valued our work.

Comment 1: Concerning the results of the preliminary experiments of localized intracellular heating, the authors

write in the manuscript that high temperature differences ($\Delta T=+5^{\circ}\text{C}$) induced cytotoxicity, which has been described as an increase in cell detachment from the dish during or after heating (lines 127-129). For instance, it would be worth showing results of viability/toxicity for the different temperature increases (+2, +3 and $+5^{\circ}\text{C}$) to support that temperature increase of $+5^{\circ}\text{C}$ is cytotoxic, whereas $\Delta T=+2^{\circ}\text{C}$ and $+3^{\circ}\text{C}$ does not induce harmful effects and represents a more physiological condition.

Response: As Reviewer #3 suggested, we have provided an index of cell viability in the Supplementary Information to show the cytotoxicity of each intensity of heating (Supplementary Figure S6).

Supplementary Figure S6. Cell viability for the different temperature increases.

Comment 2: The acquisition of a differentiated phenotype is not sufficiently supported, since the authors investigate only neurite outgrowth and the number of cells with protrusions longer than their cell bodies. The authors should at least confirm neuronal differentiation by the analysis of the expression of neuronal markers, such as for example NeuN, which is a standard marker for mature neurons.

Response: As suggested by Reviewer #3, we examined the expression of a marker of neuronal differentiation (Microtubule associated protein 2, MAP2) before and after the addition of NGF. The results showed that MAP2 expression was significantly increased in NGF-treated cells compared to untreated cells. Furthermore, MAP2 expression was increased in NGF-treated cells in which the local region of the nucleus was heated as compared to cells that were not heated. We therefore added this text and figure to the revised manuscript:

Text added to the revised manuscript (Results section): “We examined the expression of a marker of neuronal differentiation (microtubule-associated protein 2, MAP2) before and after the addition of NGF to confirm the acquisition of differentiated phenotypes. Immunostaining of cells confirmed the expression of MAP2 in NGF-treated cells (Supplementary Figure S1).” (Line 97-100 in the revised manuscript)

Supplementary Figure S1. Expression of MAP2 upon induction of neuronal differentiation by NGF. Bright field and immuno-fluorescence images of PC12 cells with or without 24 h of NGF treatment. Scale bar represents 100 μm .

Text added to the revised manuscript (Results section): “Furthermore, MAP2 expression was increased in NGF-treated cells in which local regions of the nucleus were heated under the same conditions ($\Delta T = 3\text{ }^{\circ}\text{C}$, 30 min) as compared to cells that were not heated (Supplementary Figure S7).”

Supplementary Figure S7. Influence of nuclear heating on the expression of MAP2 during neuronal differentiation by NGF. (a) Bright field and immuno-fluorescence images of PC12 cells after 24 hours with and without heating immediately after NGF addition. Scale bar represents 25 μm . (b) Quantified MAP2 levels in cells with and without heating. $**P < 0.01$ (unpaired Student's *t*-test.).

Comment 3: Referring to Figure 6, the authors write that neurite length decreases because of the administration of

the different inhibitory agents. However, it seems that cells are suffering, and I wonder if effects on neurite retraction could be due to a change in cell morphology related to the stress and maybe to high inhibitor concentrations. I would ask the authors to elucidate this issue.

Response: In fact, cycloheximide (CHX) inhibits translation, an important cellular process, and thus appears to have a broad effect on cell functions. In the observation of a large number of cells shown in Figure R1, adhesion is affected in some cells and they have a rounded shape while others have an intact shape. In this study, we analyzed cells that did not show any morphological changes such as adhesion inhibition; therefore the effect on neurite outgrowth does not appear to be a change in cell morphology related to stress. The method of this analysis is described in the Methods section.

Figure R1. The morphology of PC12 cells after 24 h of cycloheximide (CHX) treatment. Bright field images are shown. Scale bar represents 100 μm .

Text added to the revised manuscript (Methods section): “Cells that detached from the dish or displayed significant morphological alterations following microinjection, laser heating, or treatment with inhibitors were excluded from the analysis.” (Line 351-353 in the revised manuscript)

Comment 4: In the manuscript, lines 179-180, the authors write: “the temperature decrease was dependent on the intracellular compartments as inhibition of transcription affected nuclear and cytoplasmic temperatures, whereas inhibition of translation affected only the cytoplasm (Figure. 6d). However, in Figure 6D all the temperature are significantly decreased. Could the authors explain their assumption which is in contrast with the data they show?”

Response: We sincerely appreciate the point raised by Reviewer #3. The wording was our mistake—translation inhibition affected both nuclear and cytoplasmic temperatures. We have therefore revised the text as follows:

Text removed from the previous manuscript (Results section): “the temperature decrease was dependent on the intracellular compartments as inhibition of transcription affected nuclear and cytoplasmic temperatures, whereas inhibition of translation affected only the cytoplasm (Figure. 6d).”

Text added to the revised manuscript (Results section): “Transcriptional inhibition significantly affected nuclear and cytoplasmic temperatures, whereas translational inhibition affected the cytoplasm more than nuclear temperature changes (Figure 6d).” (Line 196-197 in the revised manuscript).

Comment 5: The employment of FNDs should be better explained, since the authors sometimes write that the use of FNDs is for extracellular temperature measurement, whereas in some other parts of the manuscript they write about intracellular temperature (lines 513-514).

Response: Two types of FNDs were used in this study: PEI-modified FNDs and unmodified FNDs. Unmodified FNDs have a negative charge, which makes intracellular uptake difficult, while PEI-modified FNDs have a positive charge, which makes their intracellular uptake efficiency much higher than that of unmodified FNDs. Therefore, PEI-modified FNDs were used to measure the temperature inside the cell while unmodified FNDs were used to measure the temperature on the extracellular surface. To avoid any misunderstanding, we have changed the legend text in the main text and in Supplementary Figure 9 as follows:

Text added to the revised manuscript (Results section): “By contrast, no temperature increase was observed when using unmodified FNDs positioned at the outer surface of the cells (Supplementary Figure S9)” (Line 179-180 in the revised manuscript).

Text added to the revised manuscript (Supplementary information): “Measuring the temperature of outer surface of cells during differentiation using unmodified FNDs.” (Supplementary Figure S9 legend; Line 153-154 in the revised Supplementary Information)

Comment: 6) Please, add statistical analysis for data in Figure 1b.

Response: As directed by Reviewer #3, we added a statistical analysis of the data in Figure 1b. In addition, we made the same corrections to Figure 7, which lacked statistical analysis.

Some minor revisions and typo corrections are also required through the manuscript. For example:

Comment: In line 44, “Not only is brain function,” please correct “is” with “in”.

Response: As Reviewer #3 pointed out, we have revised the manuscript.

Comment: Please, reformulate the sentence in lines 170-171.

Response: As Reviewer #3 pointed out, we have revised the manuscript as follows:

Text added to the revised manuscript (Results section): “Transcription, translation, and actin polymerization are the key intracellular reactions responsible for neuronal differentiation. When their reactions were inhibited using actinomycin D (0.8 μ M), cycloheximide (1.0 μ M), and cytochalasin D (0.5 μ M), respectively, neurite outgrowth was significantly reduced 24 h after the addition of inhibitors (Figure. 6a, b).” (Line 188-191 in the revised manuscript).

Comment: Please, correct line 517: ms = 0, \pm 1.

Response: As Reviewer #3 pointed out, we have revised the manuscript (Line 577 in the revised manuscript).

Reviewers' Comments:

Reviewer #1:

Remarks to the Author:

I am satisfied that the authors have rebutted my major concerns with their original submission, regarding the distinction between heat merely accelerating an already initiated biological process versus also having a more direct role.

The addition of the primary cortical culture experiment is welcome, though it still should not be described as *in vivo*.

Reviewer #2:

Remarks to the Author:

The manuscript was well written and reasonably revised. It can be accepted in current form.

Reviewer #3:

Remarks to the Author:

The authors have addressed all concerns and the manuscript has been improved accordingly.

[Point-by-point responses and revision in accordance with reviewer's comments]

Ms ID: NCOMMS-23-25324A

Ms title: Implication of thermal signaling in neuronal differentiation revealed by manipulation and measurement of intracellular temperature

Reviewer #1 (Remarks to the Author)

Comment: I am satisfied that the authors have rebutted my major concerns with their original submission, regarding the distinction between heat merely accelerating an already initiated biological process versus also having a more direct role.

Response: We appreciate that Reviewer #1 valued our revision.

Comment: The addition of the primary cortical culture experiment is welcome, though it still should not be described as *in vivo*

Response: As Reviewer #1 pointed out, we have removed the description of "*in vivo*" and revised as follows:

Text added to the revised manuscript (Results sections):

“Finally, we investigated the physiological significance of the contribution of intracellular temperature increase to neurite outgrowth elucidated in PC12 cells using primary mouse cortical neurons.” (Line 227-228 in the revised manuscript)